# Alcohol and the risk of pneumonia: a systematic review and meta-analysis

Evangelia Simou, John Britton, Jo Leonardi-Bee

## ABSTRACT

**Objective** A systematic review and meta-analysis to estimate the magnitude of the association between alcohol consumption and the risk of community-acquired pneumonia (CAP) in adults was undertaken.

**Design** Systematic review and meta-analysis.

**Methods** Comprehensive searches of Medline, Embase and Web of Science were carried out to identify comparative studies of the association between alcohol intake and CAP between 1985 and 2017. Reference lists were also screened. A random-effects meta-analysis was used to estimate pooled effect sizes. A dose–response meta-analysis was also performed.

**Results** We found 17 papers eligible for inclusion in the review, of which 14 provided results which could be pooled. Meta-analysis of these 14 studies identified an 83% increased risk of CAP among people who consumed alcohol or in higher amounts, relative to those who consumed no or lower amounts of alcohol, respectively (relative risk=1.83, 95% CI 1.30 to 2.57). There was substantial between-study heterogeneity, which was attributable in part to differences in study continent, adjustment for confounders and pneumonia diagnosis (clinical vs death). Dose–response analysis found that for every 10–20 g higher alcohol intake per day, there was an 8% increase in the risk of CAP.

**Conclusions** The findings suggest that alcohol consumption increases the risk of CAP. Therefore, strengthening policies to reduce alcohol intake would be likely to reduce the incidence of CAP.

## Strengths and limitations of this study

► This study represents a comprehensive review of the global literature with no language restrictions, while adhering to the guidelines of the Preferred Reporting Items for Systematic Reviews and Meta-Analyses and the Meta-analysis Of Observational Studies in Epidemiology.

► Heterogeneity was explored using subgroup analysis based on a priori defined factors.

► A dose–response analysis of alcohol consumption was also performed.

► Confounding as a result of the existence of other factors that were not usually adjusted for in the included studies (eg, socioeconomic status, malnutrition) could not be explored.

Division of Epidemiology and Public Health, UK Centre for Tobacco and Alcohol Studies, University of Nottingham, Nottingham, UK

**Correspondence to**
Miss Evangelia Simou;
evangelia.simou@nottingham.ac.uk

## INTRODUCTION

Pneumonia is a major cause of global morbidity and mortality. In 2014 in the USA, pneumonia (including influenza) was the eighth leading cause of death,[1] and according to the WHO, in 2015 pneumonia was responsible for 16% of all deaths in children aged under 5 years.[2] Community-acquired infections are the most common cause of pneumonia, and with an annual incidence in Europe and North America of between 5 and 11 cases per thousand adults,[3] community-acquired pneumonias (CAPs) account for a total of 4 million deaths annually.[4] Globally, *Streptococcus pneumoniae* is the most common pathogen causing CAP.[5] The annual incidence of CAP requiring hospitalisation among US adults is 24.8 cases per 10 000 adults, with the highest incidence especially in oldest people.[6] Patients with severe CAP admitted to European intensive care units have a mortality rate of 27% at 6 months.[7]

Pneumonia is more common with increasing age,[8 9] among people who smoke,[10–12] have low body mass index,[13] or have comorbidities including other respiratory diseases,[12 14] cardiovascular diseases,[14] stroke,[14] dementia,[11 14] and liver or renal disease.[14] Alcohol consumption is a potential risk factor for pneumonia. There are several possible mechanisms to explain the observation that alcohol consumption increases the risk of pneumonia, including the sedative properties of alcohol which can reduce oropharyngeal tone, leading to an increased risk of aspiration of microbes. Furthermore, high levels of alcohol intake can modify alveolar macrophage function, hence diminishing pulmonary defence against infection.[15 16] Also, high alcohol consumption is often associated with malnutrition[17] as it interacts with nutrient metabolism and utilisation,[18] resulting in the impairment of immunity and increased CAP risk.

To date, however, evidence on the association between alcohol consumption and CAP is limited. A systematic review and meta-analysis published in 2010, using evidence published before August 2009, found a 6% increase in

the risk of pneumonia per standard drink of 12 g of pure alcohol per day, but the number of studies reviewed (five) was small.[19] However, there is an increase in the interest on this topic, and also several studies have been published in the past 9 years. For this reason we have carried out a systematic review and meta-analysis to quantify the association between alcohol consumption and the risk of CAP.

## METHODS
The systematic review and meta-analysis was carried out in adherence to the guidelines of the Preferred Reporting Items for Systematic Reviews and Meta-Analyses[20] and the Meta-analysis Of Observational Studies in Epidemiology[21]. The protocol was published in the National Institute for Health Research international prospective register of systematic reviews (PROSPERO) under PROSPERO registration number 42015029910.

### Patient and public involvement
No patients or the public were involved in this review.

### Inclusion criteria
The Population-Exposure-Outcome-Study Design (PICO) criteria were used to determine eligibility of the articles based on the type of study design, type of population, type of exposure and outcome. We included all comparative study designs (longitudinal, cohort, case–control and cross sectional) assessing the association between alcohol intake and the risk of CAP in generally representative adult populations (≥18 years), and therefore excluded studies of selected populations such as people with HIV, hepatitis B or C virus infection, and those with hospital-acquired pneumonia. Where possible, we also analysed the association between alcohol consumption and the occurrence of pneumonia due to specific organisms (eg, *S. pneumoniae*).

### Exposure ascertainment
Alcohol consumption was defined either by self-report (interview or questionnaire) or using medical records. Also, alcohol use corresponded to drinking levels (low, moderate, heavy and alcoholism) or to frequency measures (grams/units/drinks per day/week).

### Outcome ascertainment
CAP diagnosis was based on clinical diagnosis (chest X-ray, blood test), physician diagnosis and medical records including the International Classification of Diseases (ICD) codes, or self-report.

### Search strategy
Comprehensive search strategies were applied to Medline (via Ovid), Embase (via Ovid) and Web of Science databases for the period from December 1985 to December 2017. We used search filters for observational study designs[22] and search terms for both outcome and exposure developed from relevant Cochrane Review groups.[23] When searching, medical subject heading terms were used for Medline and Embase, whereas free-text words were used for Web of Science. The Medline search filters were the following: [exp Alcohol-Related Disorders/OR Alcohol Drinking/OR (alcohol adj3 (drink$ ORor intoxicat$ OR use$ OR abus$ OR misus$ OR risk$ OR consum$ OR withdraw$ OR detox$ OR treat$ OR therap$ OR excess$ OR reduc$ OR cessation OR intervention$)).tw. OR (drink$ adj3 (excess OR heavy OR heavily OR harm OR harmful OR hazard$ OR binge OR problem$)).tw. OR alcoholic$.tw.] AND [exp Respiratory Tract Infections/OR (acute respiratory infection*.tw.) OR (lower respiratory infection*.tw.) OR (lower respiratory tract infection*.tw.) OR exp Pneumonia/OR (pneumon* OR bronchopneumon* OR pleuropneumon*).tw. OR exp Bronchitis/OR (bronchit* OR bronchiolit*).tw]. The full search strategy is presented in online supplementary table E1). The reference lists of included studies were also screened in order to identify further potentially eligible studies. No language limitation was imposed, and where necessary papers were translated into English. Where there was more than one report of findings from the same population (eg, an abstract and then a full paper), the most recently published version of the study was used. Screening of titles and abstracts, as well as the full text, was conducted independently by two reviewers (ES and JL-B). Any disagreements were resolved through discussion or with the help of a third reviewer (JB).

### Data extraction
Two reviewers (ES and JL-B) independently extracted data using a previously piloted form (see online supplementary table E2), which included the following information: author, year, study design, definitions of exposure (alcohol) and outcome (CAP), geographical location, reference population, and adjustment for confounders.

For categorical measures of alcohol drinking, where possible we compared any alcohol consumption with no alcohol consumption (reference group), or else used the lowest exposed category as the reference group. Also, in the main analysis, categorical measures of alcohol consumption were further defined as levels of consumption: light, moderate, heavy, binge and alcoholism. Grams of daily alcohol consumption were used as a standard measure, defining one drink as 0.6 ounces, 14.0 g or 1.2 tablespoons of pure alcohol.[24] Where possible, we followed the Centers for Disease Control and Prevention guidelines for the definition of heavy drinking as a weekly consumption of 15 or more drinks for men, and 8 or more drinks for women; binge drinking as 5 or more drinks during a single occasion for men, or 4 or more for women; and excessive drinking as the presence of either binge or heavy drinking.[24] The Dietary Guidelines for Americans define moderate alcohol drinking as the daily consumption of up to one drink for women and two drinks for men.[25] Otherwise we accepted the definitions of alcohol that the included studies used.

## Quality assessment

Two authors (ES and JL-B) independently assessed the methodological quality of the included studies using the Newcastle-Ottawa Scale.[26] In the process of quality assessment of each article, a maximum score of 9 stars can be obtained, whereas studies with lower quality obtain fewer stars. In case of a cohort study, the cohort study criteria were used, whereas for case–control studies the case–control criteria were used. However, for a cross-sectional study, a modified version of the case–control study criteria was used and in this case a maximum of 7 stars was given. All studies, irrespective of their design, were considered to be of high quality if they obtained a score of ≥6 stars. Discrepancies were resolved through discussion and consensus. We did not attempt to assess the methodological quality for studies published only in abstract form.

## Statistical analysis

Relative measures of risk were extracted as ORs, relative risks (RR) or HRs with 95% CIs. Where available, we used measures of risk adjusted for smoking and socioeconomic status and extracted the results separately for men and women. Where raw data were extracted from studies, we estimated ORs for case–control studies and RRs for longitudinal, cohort and cross-sectional studies. Where exposure to alcohol was reported using quantiles, or categories, we extracted adjusted effect measures relating to a comparison of the highest with the lowest exposure group.

The pooled RR and the 95% CI were estimated through pooling ORs and RRs together, since it was assumed that these two measures of effect would be similar due to the outcome measure being uncommon (prevalence <~10%).[27] However, HRs were not pooled with other effect measures. The decision to present only relative risks was made due to issues associated with using absolute risks, namely the risk difference is naturally constrained which may create difficulties when applying results to other patient groups and settings. Therefore, absolute measures are less likely to be generalisable.[28] Meta-analysis was conducted, based on the DerSimonian and Laird's random-effects model, to pool the results from the individual studies.

Heterogeneity between studies was quantified using $I^2$ statistics,[29] and explored using subgroup analyses according to study quality, study design, adjustment for confounders, alcohol reference group (no alcohol vs lowest exposed category), CAP diagnosis (clinical diagnosis vs death records), geographical location (low-income and middle-income countries vs high-income countries) and measure of effect estimated (ORs vs RRs). Funnel plots were used as a visual aid to detect publication bias, and where data for at least 10 studies were available we formally assessed publication bias using Egger's asymmetry test. We performed all analyses using Stata V.14 and Review Manager V.5.3. All p values <0.05 were deemed to represent statistical significance.

## Dose–response assessment

To assess the evidence for causality, we applied a modified version of Hill's criteria to assess causation[30] on strength of association, consistency, temporality, biological gradient and plausibility. To assess the biological gradient criterion, we performed a random-effects dose–response meta-analysis,[31 32] where we assumed a linear dose–response relation and allowed for study-level correlations across the categories of quantities of alcohol. The dose–response relation between alcohol consumption and CAP was analysed using the subgroup of studies including at least three different categories of exposure, standardised for analysis to grams per day, and where appropriate using the midpoint of categories defined by ranges of intake. If the highest exposure category was open-ended, we took the highest category midpoint to be the lower bound plus 1.2 times the lower boundary.[33] When available we included results for men and women separately.

Separate dose–response meta-analyses were conducted for cohort/longitudinal and case–control/cross-sectional studies. Dose categories relating to quantities of alcohol were created to equate to 10–20 g of pure alcohol per day (approximately one drink per day); where studies reported categories which contained the same dose ranges, we collapsed these into a single dose category through estimating a pooled effect estimates based on a fixed-effect meta-analysis model. Where necessary, effect estimates and 95% CI were back-calculated from floated to conventional CIs to enable comparisons to be made with the reference group (non-drinkers or the lowest exposed category).[34]

## RESULTS

The searches identified a total of 4589 studies published between December 1985 and December 2017, of which 17 were eligible for inclusion in the systematic review (figure 1). The characteristics of the 17 included studies are presented in table 1. A total population of 287 184 people were included in our review. Seven studies used a cohort or longitudinal design,[10 35–40] nine used a case–control design[11 41–48] and one used a cross-sectional design.[49] Eight studies were conducted in America,[10 11 39 40 46–49] five in Europe,[37 41 43–45] two in Asia[35 36] and two in Australia.[38 42] Three studies reported separate estimates of the association between alcohol and CAP for men and women,[10 41 44] and 12 studies reported effect estimates adjusted for confounders.[10 35 36 39 41 43–49]

The majority of studies assessed alcohol consumption by self-report, based either on a standardised questionnaire or on an interview, while five studies used reported intake data from medical records.[11 37 40 46 47] The reference group for nine studies comprised people who never consumed alcohol,[10 35 36 38 42 44 45 47 48] whereas the reference group for the remaining eight studies comprised people who consumed the lowest quantity of alcohol.[11 37 39–41 43 46 49]

Seven studies ascertained CAP using a clinical diagnosis, and five of these used chest X-ray

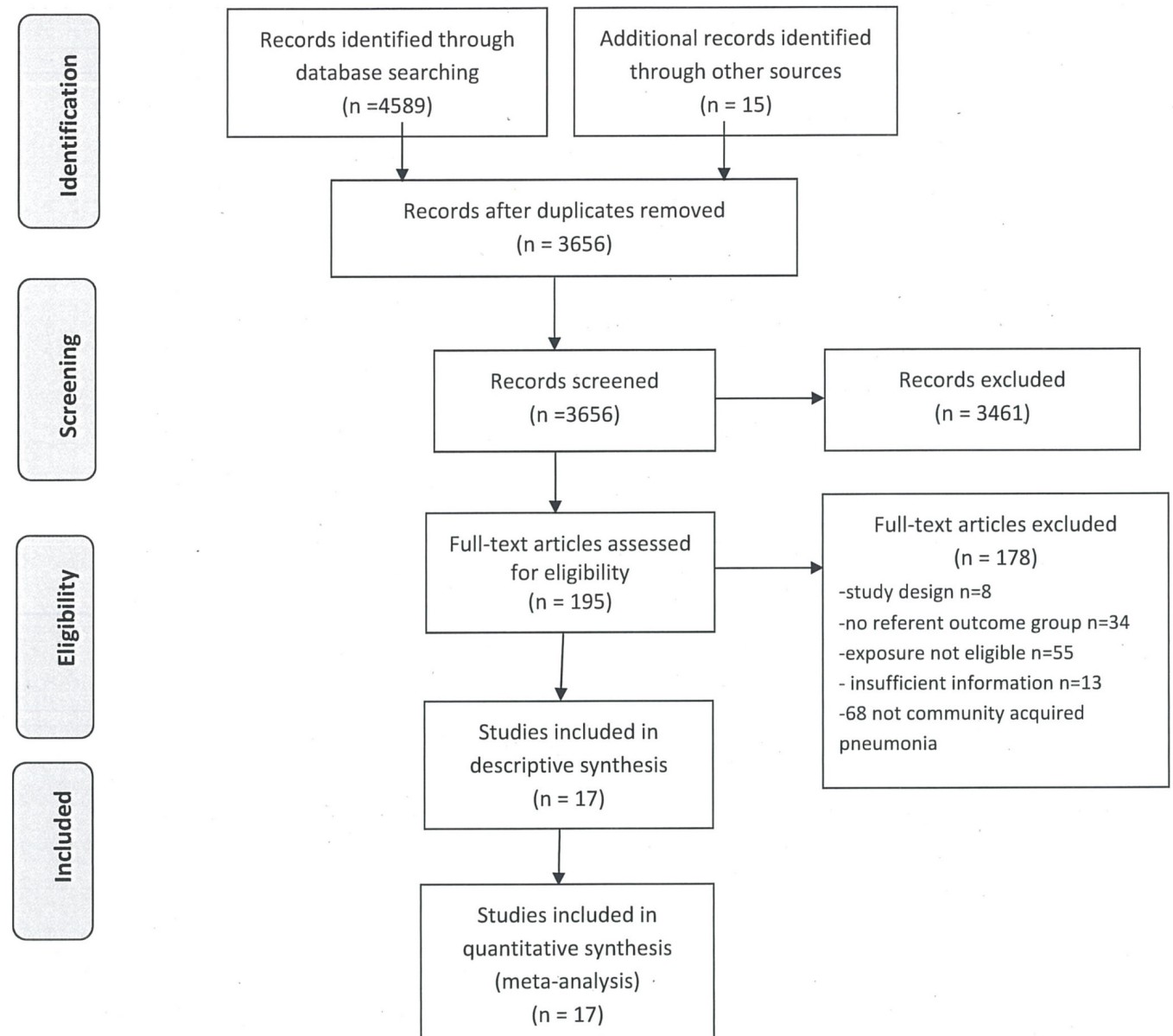

**Figure 1** Study selection.

radiography.[42–45 48] A further seven studies ascertained CAP using ICD codes[35 36 38 40 41 46] and medical records,[46] and two studies used self-report interviews.[39 49] The remaining study ascertained CAP via physician diagnosis using medical records.[10]

The methodological quality of the case–control, cohort and cross-sectional studies ranged from 5 to 8, with a median score of 6. Ten studies were deemed to be of high quality (>6 score),[10 35 37–39 41 43 45–47] whereas lower scores tended to arise from failure to adjust for confounders or using self-reported methods to ascertain alcohol consumption. The results of the quality assessment are presented in detail in table 2.

### Meta-analysis findings

Fourteen of the 17 included studies provided data from which pooled RRs could be estimated, and a pooled

analysis of these studies found the risk of CAP to be significantly increased in people who consumed alcohol at all or in higher amounts, relative to those who consumed no or lower amounts of alcohol, respectively (pooled RR=1.83, 95% CI 1.30 to 2.57, $I^2$=91%; figure 2). There was no evidence of publication bias detected visually via a funnel plot (see online supplementary figure E1) and statistically via Egger's asymmetry test (p=0.596).

Subgroup analyses exploring the reason for heterogeneity in the meta-analysis of these 14 studies are presented in online supplementary table E3). Heterogeneity was not explained by study design (case–control, longitudinal/cohort, cross-sectional; p for subgroup differences=0.07), methodological quality (high vs low; p=0.09) or gender (male vs female; p=0.74). However, significant differences were found according to adjustment for

**Table 1** Characteristics of the included studies

| Study and year | Study design | Geographical location | Alcohol ascertainment | Alcohol definition | CAP ascertainment | Confounders adjusted | Effect estimate |
|---|---|---|---|---|---|---|---|
| Almirall et al,[45] 1999 | Case–control | Europe/Spain | Self-report/questionnaire | Quartiles of alcohol intake >35.3 vs 0 (g/day) | Clinically suspected and chest radiography | Age, sex, municipality | OR |
| Almirall et al,[44] 2008 | Case–control | Europe/Spain | Self-report/questionnaire | Quartiles of alcohol intake (g/day) Men: >80 vs 0 Women: >40 vs 0 | Clinically suspected and chest radiography | Age, sex, primary care practice | OR |
| Baik et al,[10] 2000 | Cohort | America/USA | Self-report/questionnaire | Men: >30 vs never Women: >30 vs never (g/day) | Physician diagnosis/medical records | Age, smoking status, BMI, quintile of metabolic equivalent | Relative risk |
| Breitling et al,[39] 2016 | Cohort | America/USA | Self-report/questionnaire | Quartiles of alcohol intake (g/day) Men: >20 vs ≤20 Women: >10 vs 0 | Self-report questionnaire | Age, sex, smoking, BMI, diabetes mellitus, stroke, congestive heart failure, cancer | Relative risk |
| Clough et al,[42] 2003 | Case–control | Australia | Self-report/interview | Alcohol yes vs alcohol no | Clinically suspected/X-ray findings | –* | OR |
| Fernández-Solá et al,[43] 1995 | Case–control | Europe/Spain | Self-report/interview and questionnaire | High intake (men: >100g, women: >80g) vs lower intake (g/day 2 years before submission) | Clinically suspected/chest X-ray | Liver cirrhosis, smoking, COPD, diabetes, heart failure, malnutrition | OR |
| Inoue et al,[36] 2007 | Cohort | Asia/Japan | Self-report/questionnaire | Current vs never drinking | Mortality ICD codes | Age and history of diabetes mellitus | HR |
| Jackson et al,[46] 2009 | Case–control | America/USA | Medical records | Current alcoholism vs no alcoholism | ICD-9 codes | Age, sex, pneumonia-free person-time | OR |
| Koivula et al,[37] 1994 | Cohort | Europe/Finland | Medical records | Alcoholism vs no alcoholism | Medical records | Age, sex, chronic conditions | Relative risk |
| Lipsky et al,[11] 1986 | Case–control | America/USA | Medical records | Heavy vs moderate (drinks/day) | Clinically suspected | –* | Relative risk |
| Loeb et al,[48] 2009 | Case–control | America/USA | Self-report/questionnaire | Alcohol yes (previous 12 months) vs alcohol no (g/month) | Clinically suspected and chest radiography | Multivitamins, smoking, history of gas and fumes exposure | OR |
| Phung and Wang,[38] 2013 | Cohort | Australia | Self-report/questionnaire | Alcohol yes vs alcohol no | Hospital records—ICD codes | –* | HR |
| Quraishi et al,[49] 2013 | Cross-sectional | America/USA | Self-report/interview | Alcohol consumption (≤30 vs >30 drinks per month) | Self-report interview | –* | Relative risk |

Continued

**Table 1** Continued

| Study and year | Study design | Geographical location | Alcohol ascertainment | Alcohol definition | CAP ascertainment | Confounders adjusted | Effect estimate |
|---|---|---|---|---|---|---|---|
| Shen et al,[35] 2013 | Cohort | Asia/China | Self-report/interview | Excessive vs never drinkers (units/week) | Mortality ICD codes | Age, sex, education, housing, monthly expenditure, smoking, BMI, exercise, health status | HR |
| Watt et al,[47] 2007 | Case–control | America/USA | Medical records | Alcoholism/alcohol use vs no use of alcohol | Clinically suspected pneumococcal isolation in patient from sterile body fluid | Smoking, BMI, electricity/indoor plumbing in home, living with unvaccinated child, unemployed, wood/coal, smoke | OR |
| Yende et al,[40] 2013 | Cohort | America/USA | Medical records | Alcohol abuse vs no alcohol abuse | ICD-9 codes | –* | Relative risk |
| Zaridze et al,[41] 2009 | Case–control | Europe/Russia | Self-report interview | ≥3 bottles(per week) vs ≤0.5 bottles of vodka | ICD codes Death records | Age, city and smoking | Relative risk |

*Crude analysis reported.

BMI, body mass index; CAP, community-acquired pneumonia; COPD, chronic obstructive pulmonary disease; ICD, International Classification of Diseases.

**Table 2** Quality assessment: Newcastle-Ottawa Scale

| Study, year | Number of stars | | | |
| --- | --- | --- | --- | --- |
| | Selection* | Comparability† | Exposure‡ | Overall |
| Almirall et al,[45] 1999 | 4 | 1 | 1 | 6/9 |
| Almirall et al,[44] 2008 | 3 | 1 | 1 | 5/9 |
| Baik et al,[10] 2000 | 4 | 2 | 2 | 8/9 |
| Breitling et al,[39] 2016 | 3 | 2 | 2 | 5/9 |
| Clough et al,[42] 2003 | 4 | 0 | 1 | 5/9 |
| Fernández-Solá et al,[43] 1995 | 3 | 2 | 1 | 6/9 |
| Inoue et al,[36] 2007 | 3 | 1 | 1 | 5/9 |
| Jackson et al,[46] 2009 | 4 | 1 | 1 | 6/9 |
| Koivula et al,[37] 1994 | 4 | 1 | 3 | 8/9 |
| Lipsky et al,[11] 1986 | 3 | 0 | 2 | 5/9 |
| Loeb et al,[48] 2009 | 2 | 2 | 1 | 5/9 |
| Phung and Wang,[38] 2013 | 3 | 0 | 3 | 6/9 |
| Quraishi et al,[49] 2013 | 1 | 0 | 1 | 2/6 |
| Shen et al,[35] 2013 | 3 | 2 | 3 | 8/9 |
| Watt et al,[47] 2007 | 4 | 2 | 1 | 7/9 |
| Yende et al,[40] 2013 | 4 | 0 | 2 | 6/9 |
| Zaridze et al,[41] 2009 | 3 | 2 | 1 | 6/9 |

*Maximum 4 stars.
†Maximum 2 stars.
‡Maximum 3 stars.

confounders (adjusted vs unadjusted; p=0.03), continent of study (America, Europe, Australia; p=0.0003), and ascertainment of CAP (clinical diagnosis vs death records; p=0.002). Furthermore no difference was found for studies that presented OR estimates compared with studies that presented RR estimates (p for subgroup differences=1.00).

Additionally, no significant differences were found by the definition of the reference group for alcohol consumption (p=0.39; figure 2). However, high heterogeneity ($I^2$=95%) was detected within the second subgrouping which used the lowest category of exposure as the reference group, where the following definitions were used: no alcoholism,[37 46] no alcohol abuse,[40] moderate drinking,[11] ≤30 drinks/month,[49] ≤0.5 bottles of vodka,[41] <100 g/day for men and <80 g/day for women,[43] and <20 g/day and <10 g/day for men and women, respectively[39]; however, the gradient of exposure did not seem to be related to the magnitude of effect.

A sensitivity analysis restricted to the six studies which provided smoking-adjusted estimates found a larger magnitude of effect compared with the main analysis (pooled RR=2.01, 95% CI 1.25 to 3.23, $I^2$=93%, 6 studies). Similarly the studies that provided age-adjusted effect estimates found a risk of 1.90 (pooled RR=1.90, 95% CI 1.20 to 3.02, $I^2$=93%, 7 studies).

The remaining three studies presented effect estimates as HRs,[35 36 38] and a pooled analysis of these studies estimated an HR for CAP in relation to alcohol consumption of 0.90 (pooled HR=0.90, 95% CI 0.79 to 1.03, $I^2$=0, 3 studies).

Two studies assessing the effect of alcohol on pneumococcal disease-specific strains of pneumonia were identified.[11 47] A pooled analysis of these studies found that there was more than a doubling of risk of *S. pneumoniae* CAP in people who consumed alcohol (RR=2.16, 95% CI 1.05 to 4.48, $I^2$=42%).

### Biological gradient meta-analysis

Five of the included studies provided data enabling a dose–response meta-analysis,[10 41 42 44 45] of which one used a cohort design (data reported separately for men and women) and four were case–control studies. A pooled analysis of the dose–response data from the cohort study found no significant gradient in the quantity of alcohol associated with the risk of CAP (p for trend=0.136). In contrast, the pooled analysis of the dose–response data from the four case–control studies indicated that there was a significant gradient in the quantity of alcohol associated with an 8% increase in the risk of CAP for every 10–20 g of pure alcohol consumed per day (equivalent to 1 drink/day) (pooled RR=1.08, 95% CI 1.06 to 1.09, p<0.0001; figure 3).

### DISCUSSION

Alcohol consumption is a recognised and avoidable risk factor for a range of diseases and injuries, including

| Study or Subgroup | log[Relative risk] | SE | Weight | Relative risk IV, Random, 95% CI | Relative risk IV, Random, 95% CI |
|---|---|---|---|---|---|
| **12.1.1 alcohol vs no alcohol** | | | | | |
| Almirall 1999 | 0.1989 | 0.3257 | 5.8% | 1.22 [0.64, 2.31] | |
| Almirall men 2008 | 0.8502 | 0.3719 | 5.5% | 2.34 [1.13, 4.85] | |
| Almirall women 2008 | -0.2231 | 0.6727 | 3.5% | 0.80 [0.21, 2.99] | |
| Baik men 2000 | 0.157 | 0.2112 | 6.6% | 1.17 [0.77, 1.77] | |
| Baik women 2000 | 0.157 | 0.5118 | 4.5% | 1.17 [0.43, 3.19] | |
| Clough 2003 | 0.6678 | 0.3028 | 6.0% | 1.95 [1.08, 3.53] | |
| Loeb 2009 | 0.5247 | 0.2217 | 6.6% | 1.69 [1.09, 2.61] | |
| Watt 2007 | 1.0647 | 0.3172 | 5.9% | 2.90 [1.56, 5.40] | |
| **Subtotal (95% CI)** | | | **44.4%** | **1.61 [1.25, 2.08]** | |
| Heterogeneity: Tau² = 0.03; Chi² = 9.37, df = 7 (P = 0.23); I² = 25% | | | | | |
| Test for overall effect: Z = 3.67 (P = 0.0002) | | | | | |
| | | | | | |
| **12.1.2 alcohol vs lowest exposure category** | | | | | |
| Breitling 2016 | -0.123 | 0.134 | 7.0% | 0.88 [0.68, 1.15] | |
| Fernández-Solá 2007 | 1.6544 | 0.6635 | 3.5% | 5.23 [1.42, 19.20] | |
| Jackson 2009 | 0.3365 | 0.2533 | 6.4% | 1.40 [0.85, 2.30] | |
| Koivula 1994 | 2.1972 | 0.2999 | 6.0% | 9.00 [5.00, 16.20] | |
| Lipsky 1986 | 0.3008 | 0.4871 | 4.7% | 1.35 [0.52, 3.51] | |
| Quraishi 2013 | 0.1782 | 0.2221 | 6.6% | 1.20 [0.77, 1.85] | |
| Yende 2013 | -0.0305 | 0.1128 | 7.1% | 0.97 [0.78, 1.21] | |
| Zaridze men 2009 | 1.1916 | 0.0772 | 7.2% | 3.29 [2.83, 3.83] | |
| Zaridze women 2009 | 1.2296 | 0.1332 | 7.0% | 3.42 [2.63, 4.44] | |
| **Subtotal (95% CI)** | | | **55.6%** | **2.07 [1.24, 3.44]** | |
| Heterogeneity: Tau² = 0.52; Chi² = 171.02, df = 8 (P < 0.00001); I² = 95% | | | | | |
| Test for overall effect: Z = 2.80 (P = 0.005) | | | | | |
| | | | | | |
| **Total (95% CI)** | | | **100.0%** | **1.83 [1.30, 2.57]** | |
| Heterogeneity: Tau² = 0.41; Chi² = 185.16, df = 16 (P < 0.00001); I² = 91% | | | | | |
| Test for overall effect: Z = 3.49 (P = 0.0005) | | | | | |
| Test for subgroup differences: Chi² = 0.74, df = 1 (P = 0.39), I² = 0% | | | | | |

0.01  0.1  1  10  100
decreased risk for CAP   increased risk for CAP

**Figure 2** Forest plot of alcohol consumption and risk of community-acquired pneumonia (CAP): subgroup analysis based on reference group (never drinking vs lowest drinking category).

neuropsychiatric conditions, gastrointestinal and cardio-vascular diseases, cancer, suicide, violence and tuberculosis.[50] To date, however, the association between alcohol consumption and pneumonia risk has attracted relatively little attention.

## Summary of the findings

This meta-analysis of studies published over the past 30 years demonstrates a clear and statistically significant

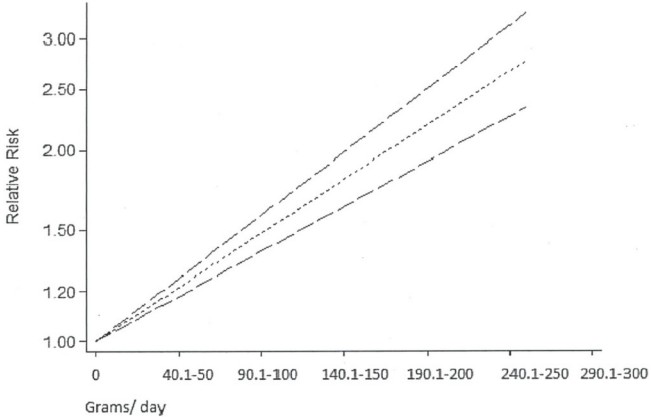

**Figure 3** Linear dose–response meta-analysis for the association between alcohol intake categories (g/day) and the risk of community-acquired pneumonia.

relation between alcohol consumption and the risk of CAP. The effect was strong, with a 1.8-fold increase in risk among those who consumed alcohol at all or in higher amounts, relative to those who consumed no or lower amounts of alcohol, respectively, and significantly related to the level of intake, with no evidence of publication bias. The dose–response analysis indicated that consuming drinks that contain 10–20 g of alcohol per day was linked to an 8% increased risk of acquiring CAP. Furthermore, the findings of the subgroup analysis indicated significant differences in the risk of pneumonia according to continent of the study, with Europe having the highest rate (threefold) for CAP risk.

## Strengths and limitations

This study represents a comprehensive review of the global literature with no language restriction, making this analysis the most complete to date and our findings likely to be generalisable. There was significant heterogeneity between the studies in our analysis, but our subgroup analyses indicate that this arose primarily from the continent in which the study was carried out (America, Europe, Australia), adjustment for confounders and the ascertainment of CAP (death vs clinical diagnosis). Misclassification bias arising from inclusion of non-drinkers in the lowest category of alcohol intake in some studies can be

another possible limitation in our review, but will result in a more conservative estimate of effect. A dose–response relationship was identified. However the included studies did not report dose–response relations separately for men and women, so we are unable to carry out a comparative analysis. Furthermore, confounding as a result of the existence of other factors that were not usually adjusted for in the included studies (eg, socioeconomic status, malnutrition) could not be explored.

## Comparison with other studies

Our findings extend those of an earlier review and meta-analysis carried out in 2010.[19] Another review focused on the risk factors for invasive pneumococcal diseases indicated an elevated risk for invasive pneumococcal disease due to alcohol consumption in six of the four studies included in the meta-analysis model.[51] Likewise, another recent meta-analysis indicated an elevated risk for invasive pneumococcal disease due to alcohol consumption in six of the four studies included in the meta-analysis model.[52] Similarly our separate meta-analysis focused on pneumococcal infections including two of these studies, due to our eligibility criteria, showed an elevated risk for pneumococcal acquisition.

A previous systematic review and meta-analysis found that people with a daily alcohol consumption of either 24 g, 60 g and 120 g have a 12%, 33% and 76% increased risk of CAP, respectively.[19] Our dose–response analysis generated a slightly less strong effect, of an 8% increase in risk per 10–20 g of (pure) alcohol consumed per day.

A general systematic review published by Almirall et al in 2017[53] focused on the risk factors of CAP, but provided only a narrative summary of findings and stating that no definite conclusion could be drawn. In contrast, our review found evidence of a doubling in the risk of CAP in people who consumed alcohol. Furthermore, our demonstration of a significant exposure–response association increases the likelihood, given the strength of the observed association and its consistency across a range of subgroups, that the observed association is causal. Further evidence of causality arises from studies demonstrating that alcohol consumption impairs alveolar macrophages and increases carriage of pneumonia pathogens.[15 16 54]

## Clinical implications

The findings from the present review highlight the need to address high alcohol consumption as a means to prevent CAP. Clinicians managing patients with pneumonia could, for example, counsel reducing alcohol intake as a means to prevent further episodes, and those addressing high alcohol consumption in more general terms could add an increased risk of pneumonia as a further reason to reduce intake.

Our findings also have implications for public health: in Europe, for example, the estimated annual cost of CAP is approximately €10.1 billion[55] and might be reduced substantially by more proactive clinical and public health measures to reduce alcohol consumption.

## CONCLUSION

Our findings thus provide clear evidence that alcohol increases the risk of pneumonia. Informing people who drink alcohol of this risk, especially those who consume high levels of alcohol, both in clinical contacts and through public health policy, may therefore help to prevent this disease.

**Acknowledgements** The authors thank Erica Brasil, Magdalena Opazo-Breton and Yue Huang from the University of Nottingham for their help in translations.

**Contributors** ES, JB and JL-B designed the study and wrote the protocol. ES wrote the search strategy and undertook the literature searches, and wrote the draft of the manuscript. ES and JL-B undertook study screening, data extraction and quality assessment. ES undertook all data analyses, supervised by JL-B. All authors contributed to the interpretation of the findings. JB and JL-B provided critical revisions to the article, and all authors approved the final version of the article to be published. ES acts as guarantor of the manuscript.

**Funding** This work was supported by the Medical Research Council (grant number MR/K023195/1). The UK Centre for Tobacco and Alcohol Studies (http://www.ukctas.net), and the British Heart Foundation, Cancer Research UK, the Economic and Social Research Council, and the National Institute for Health Research, under the auspices of the UK Clinical Research Collaboration, are gratefully acknowledged.

**Competing interests** None declared.

**Patient consent** Not required.

**Ethics approval** Ethics approval was not required for this work.

**Provenance and peer review** Not commissioned; externally peer reviewed.

**Data sharing statement** No data sharing available.

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
