## [Reviewer comments · BMJ Open]

ARTICLE DETAILS

TITLE (PROVISIONAL)	Alcohol and the risk of pneumonia: A systematic review and meta-analysis
AUTHORS	Simou, Evangelia; Britton, John; Leonardi-Bee, Jo

VERSION 1 – REVIEW

REVIEWER	Francisco Sanz Herrero Pulmonology Department, Consorci Hospital General Universitari de València, València (Spain)
REVIEW RETURNED	27-Feb-2018

GENERAL COMMENTS	INTRODUCTION The authors make a correct introduction with an adequate update of the current state of the topic. They also establish an adequate justification of the need to review the use of alcohol as a risk factor in pneumonia. Only note that when it appears in the name of bacteria (for example Streptococcus pneumoniae) it must be in italics. METHODS The authors use a correct and systematic methodology according to the established protocols for systematic reviews. The search of the published works is exhaustive and does not limit only to publications in the English language. It is noteworthy the effort made to define the degree of exposure to alcohol as it is a variable with great variability between the different studies. RESULTS The results are correctly summarized and well expressed in the tables and figures. Page 7, lines 28-31. Meta-analysis findings section. The authors find that the risk of NAC is significantly increased in people who consume alcohol in low and high amounts. The concepts of high and low quantity should be well defined. Page 8, line3 The text states: "... more than a doubling of risk of Streptococcus CAP..." It should be say: "...more than a doubling of risk of Streptococcus pneumoniae CAP..." DISCUSSION Strengths and limitation The authors should recognize as a limitation the existence of other possible confounders or cofactors that are not usually registered in studies, such as socioeconomic status, the presence of malnutrition, etc. CONCLUSION Authors stated that "...measures to reduce alcohol intake are likely to reduce mortality and morbidity from community-acquired
---

	pneumonia..." (page 9, lines 29-31), however, the aims of the study is to evaluate alcohol consumption and risk of CAP, nor its impact on the outcomes. TABLES, E-TABLES AND FIGURES The tables and figures are appropriate and summarize the results well.
--	---

REVIEWER	Vijay Sivaraman North Carolina Central University, USA
REVIEW RETURNED	08-Mar-2018

GENERAL COMMENTS	I did not observe limitations to methods of assessment, which should be included.
---

REVIEWER	Lifeng Lin Florida State University
REVIEW RETURNED	17-Apr-2018

GENERAL COMMENTS	This manuscript conducted a meta-analysis to investigate the effect of alcohol consumption in pneumonia. I think it was well written, and the analyses and the results were clearly presented. I have focused on reviewing the statistical analyses. My comments are as follows. At the top of page 4, the authors mentioned that the collected studies reported different measures of effect (including OR, RR and HR). The ORs and RRs were pooled together to get a summary RR, but HR was not pooled with other measures. The authors may provide a little more details about how the ORs and RRs were pooled together. On page 7, the authors assessed publication bias of the meta-analysis using a funnel plot, but the plot was not presented. It may be presented in the supplementary materials. Also, the authors claimed that "there was evidence of publication bias" via the funnel plot, but the P-value from the Egger's regression test was 0.596. These seem to be contradictory. In Table 1, the author may add a column that indicates the type of effect size reported by each study (OR, RR, or HR?) The authors presented only the funnel plot based on two subgroups of alcohol consumption (alcohol vs no alcohol or alcohol vs lowest category of exposure) in Figure 2. The funnel plots based on other classifications of subgroups may be presented in the supplementary materials. From Figure 2, it seems that the studies comparing alcohol vs. no alcohol were fairly homogeneous, while those comparing alcohol vs lowest category of exposure were fairly heterogeneous. The authors may investigate how differently was the "lowest category of exposure" defined in each study, because heterogeneity is a critical problem in meta-analyses and most results in this meta-analysis were highly heterogeneous (I^2 around 90%). Minor comments: On page 4, change "I^2 statistics" to "I^2 statistic".
--

	In Figure 3, please specify what are the dashed lines in the figure title. Also, the ranges on the x-axis in this plot are a bit confusing. Does each range (e.g., 40.1-50) indicate a single point on the x-axis or an interval from 40.1 to 50?
--	---

REVIEWER	Tea Kristiane Espeland Uggen University of Technology Sydney, Australia
-----------------	--

REVIEW RETURNED	23-Apr-2018
-------------

GENERAL COMMENTS	Thank you for the opportunity to review this article. Please see my comments below justifying any 'no' answers from the review checklist: Overall You have an interesting and important research question. However, several sections of your paper are very confusing and difficult to follow. Some of the statistical tests you have performed are not explained, justified or results interpreted (only listed). There is also one occasion of contradicting findings. There are also quite a few typos and inconsistencies that need to be corrected. Language, typos and inconsistencies There are several typos and inconsistencies throughout the paper:  - Page 1/line 55: Full stop between last word in sentence and reference instead of space; - Page 3/lines 3-23: Check inconsistent spacing and notation in paragraph; - Page 3/line 39: The number "8" is written in numerical form here but in letters further down the paper. This should be kept consistent; - Page 3/line 47: Missing space after full stop; - Page 4/line 16: Remove space after opening bracket; - Page 4/line 47: Missing closing bracket; - Page 5/line 12: Missing space before opening bracket; - Page 7/line 52-53: Inconsistent reporting of CI in this sentence compared to previous CI reporting on the same page; - Page 7/line 33: If this is a p-value you are reporting, keep notation consistent with other p-
---

values in your article: lower case “p”;

- Page 8/line 46: The sentence containing “... but will resulted in...” should be changed to “but would have resulted in” or “but will result in”;
- Page 8/line 49: “...did not reported...” should be “...did not report...” or “...had not reported...”;
- Page 9/line 12: Misplaced comma between “12%” and “23%”;
- Notation of decimal points keeps changing throughout the paper (x.x vs. x·x);

Methods

It's not clear which form you are referring to when stating “Two reviewers (ES and JL-B) independently extracted data using a previously piloted form...”. Do you have a reference for this? *(page 3/line 26)*

The statement “According to CDC guidelines, we defined heavy drinking as...” is confusing – did you define heavy drinking or were they already defined in the CDC guidelines. *(page 3/line38)*

You have defined levels of consumption categorically on page3/lines 34-35 as "light, moderate, heavy and alcoholism", yet on page 3/lines 39-42 the term “binge drinking” is defined in quantitative terms even though this term was not previously included as one of the four categories of consumption. Furthermore, the categories “light” and “alcoholism” are not defined quantitatively. There seems to be a lack of consistency in defining alcohol consumption. Additionally, it seems odd that “binge drinking” and “heavy drinking” should be categorised in the same group labelled “excessive drinking”. What is the reasoning behind this? Could a person not be an infrequent consumer of alcohol but still be binge drinking on rare occasions? This section is a bit unclear. *(page 3/lines 31-43)*

In the sentence: "A score of 6, or more..." - was this cut-off value used for both cohort and case control studies (which you previously mentioned was rated out of 9) and the cross sectional studies (rated out of 7)? *(page 3/line 49)*

In the sentence: "...where the outcome measure was not assumed to be common" - what is meant by “common” in this context? This sentence is a bit confusing. *(page 4/lines 8-10)*

	Results You have listed references to the nine studies which comprised people that never consumed alcohol but not for the remaining 8 studies comprised of people that consumed the lowest quantity of alcohol. (page 5/lines 13-14) In the sentence: "...quality of the included studies ranged from five to eight, with a median score of six." – are these scores out of 7 or 9? You mentioned in a previous section that some were out of 7 and some out of 9. You may want to mention what you are comparing these scores to again. (page 5/lines 21-22) The overall I^2 is very high (91%) and you should provide interpretations of the effects of this. (page 7/line 32) There are contradicting findings with the statement in "RESULTS": "There was evidence of publication bias detected visually via a funnel plot, and statistically via Egger's asymmetry test ($P=0.596$)" on page 7/line 32-33 and with the statement in "DISCUSSION": "...with no evidence of publication bias" on page 8/line 34-35. Furthermore, the p-value listed seems to be inconsistent with the conclusion drawn in the results. (page 7/lines 32-33) You should provide the funnel plot described on page 7/line 32-33 as "There was evidence of publication bias detected visually via a funnel plot, and statistically via Egger's asymmetry test". (page 7/lines 32-33) Discussion What is meant by "relatively high intakes"? You previously categorised alcohol consumption to be "low/moderate/high/alcoholism". Which ones are "relatively high"? Could you also provide a quantitative measure for this term? (page 8/lines 33-34) The concluding statement is a bit of an over-reach given not only the term "relatively high intakes of alcohol" and the high I^2 but also the conclusion drawn on morbidity and mortality. (page 9/lines 28-31)
--	---

VERSION 1 – AUTHOR RESPONSE

Comments from the Associate Editor:

This appears well written, and clearly presented.

The Discussion is very short; they should at the very least discuss the clinical implications of their findings

Response

We thank the Associate Editor for the comments.

We now have added the following text on the clinical implications of our findings in the discussion:

Clinical implications

The findings from the present review highlight the need to address high alcohol consumption as a means to prevent community acquired pneumonia. Clinicians managing patients with pneumonia could for example counsel reducing alcohol intake as a means to prevent further episodes; and those addressing high alcohol consumption in more general terms could add an increased risk of pneumonia as a further reason to reduce intake. Our findings also have implications for public health: in Europe for example, the estimated annual costs of CAP are approximately €10.1 billion (53), might be reduced substantially by more pro-active clinical and public health measures to reduce alcohol consumption.

We also have added the following text in the summary of the findings: The dose response analysis indicated that consuming drinks that contain 10-20 grams of alcohol per day was linked to an 8% increased risk of acquiring community acquired pneumonia. Furthermore, the findings of the subgroup analysis indicated significant differences in the risk of pneumonia according to continent of the study; with Europe having the highest rate (threefold) for CAP risk.

Editorial Requirements:

- Please revise the Strengths and Limitations section (after the abstract) to focus on the methodological strengths and limitations of your study rather than summarizing the results.
- Please include the original protocol for the study, if one exists, as a supplementary file.

Response

-The strengths and limitations section has been revised.

-The original protocol of the review has been submitted as a supplementary file.

Reviewer: 1

Reviewer Name: Francisco Sanz Herrero

Institution and Country: Pulmonology Department, Consorci Hospital General Universitari de València, València (Spain) Please state any competing interests: None declared

INTRODUCTION

Comment: The authors make a correct introduction with an adequate update of the current state of the topic. They also establish an adequate justification of the need to review the use of alcohol as a risk factor in pneumonia.

Only note that when it appears in the name of bacteria (for example *Streptococcus pneumoniae*) it must be in italics.

Response

We thank the reviewer for being happy with the introduction section. All names of bacteria are now written in italics.

METHODS

Comment: The authors use a correct and systematic methodology according to the established protocols for systematic reviews. The search of the published works is exhaustive and does not limit only to publications in the English language. It is noteworthy the effort made to define the degree of exposure to alcohol as it is a variable with great variability between the different studies.

Response

We thank the reviewer for the comments and also for appreciating the effort we made to define the degree of exposure to alcohol due to the high variability of this variable.

RESULTS

Comment: The results are correctly summarized and well expressed in the tables and figures. Page 7, lines 28-31. Meta-analysis findings section.

The authors find that the risk of NAC is significantly increased in people who consume alcohol in low and high amounts. The concepts of high and low quantity should be well defined.

Response

We thank the reviewer for the comment. The included studies used variable definitions for high and low alcohol consumption. To provide some consistency across studies, where possible, we used defined consumption of alcohol based on the CDC guidelines and the Dietary Guidelines for Americans, however in several papers there was insufficient information presented, therefore we used the paper's definition of high and low alcohol consumption. To clarify this, we have amended the text in the "Data extraction" section to the following: "Where possible, we followed the CDC guidelines for the definition of heavy drinking as a weekly consumption of 15, or more drinks for men, and 8 or more drinks for women; binge drinking as 5, or more drinks during a single occasion for men, or 4 or more for women; and excessive drinking as the presence of either binge or heavy drinking (24). The Dietary Guidelines for Americans defines moderate alcohol drinking as the daily consumption of up to one drink for women and two drinks for men (25). Otherwise we accepted the definitions of alcohol that the included studies used".

Comment: Page 8, line3

The text states: "... more than a doubling of risk of Streptococcus CAP..."

It should be say: "...more than a doubling of risk of Streptococcus pneumoniae CAP..."

Response

The change has been made as requested. We have also italicised "*Streptococcus pneumoniae*".

DISCUSSION

Comment: Strengths and limitation

The authors should recognize as a limitation the existence of other possible confounders or cofactors that are not usually registered in studies, such as socioeconomic status, the presence of malnutrition, etc.

Response

We thank the reviewer for the comment. We have added the following: "Furthermore, confounding as a result of the existence of other factors that were not usually adjusted for in the included studies (e.g. socioeconomic status, malnutrition) could not be explored," to the Strengths and Limitations sections of the Discussion and following the abstract.

CONCLUSION

Comment: Authors stated that "...measures to reduce alcohol intake are likely to reduce mortality and morbidity from community-acquired pneumonia..." (page 9, lines 29-31), however, the aims of the study is to evaluate alcohol consumption and risk of CAP, nor its impact on the outcomes.

Response

We thank the reviewer for this comment. The conclusion statement drawn on morbidity and mortality has now been deleted. We have added the following: Our findings thus provide clear evidence that alcohol increases the risk of pneumonia. Informing people who drink alcohol of this risk, especially those who consume high levels of alcohol, both in clinical contacts and through public health policy, may therefore help to prevent this disease.

Comment: TABLES, E-TABLES AND FIGURES

The tables and figures are appropriate and summarize the results well.

Response

We thank the reviewer for being happy with the presentation of the results in the Tables, e-tables and figures.

Reviewer: 2

Reviewer Name: Vijay Sivaraman

Institution and Country: North Carolina Central University, USA Please state any competing interests: non declared

Please leave your comments for the authors below I did not observe limitations to methods of assessment, which should be included.

Response

We thank the reviewer for being happy with the methodology of our manuscript.

Reviewer: 3

Reviewer Name: Lifeng Lin

Institution and Country: Florida State University Please state any competing interests: None declared.

Please leave your comments for the authors below

This manuscript conducted a meta-analysis to investigate the effect of alcohol consumption in pneumonia. I think it was well written, and the analyses and the results were clearly presented. I have focused on reviewing the statistical analyses. My comments are as follows.

Comment: At the top of page 4, the authors mentioned that the collected studies reported different measures of effect (including OR, RR and HR). The ORs and RRs were pooled together to get a summary RR, but HR was not pooled with other measures. The authors may provide a little more details about how the ORs and RRs were pooled together.

Response

We thank the reviewer for the comment. We have added the following in the statistical analysis: "The pooled relative risk and the 95% CI were estimated through pooling ORs and RRs together, since it was assumed that these two measures of effect would be similar due to the outcome measure being uncommon (prevalence < ~10%) (27)". However, we share the reviewer's concerns and therefore have included a subgroup analysis to assess whether there are differences according to effect estimate. "Heterogeneity between studies was quantified using I^2 statistics (27); and explored using subgroup analyses according to study quality, study design, adjustment for confounders, alcohol reference group (no alcohol vs lowest exposed category), CAP diagnosis (clinical diagnosis vs death records), geographical location (Low and Middle Income Countries versus High Income Countries) **and measure of effect estimated (ORs vs RRs)**". We have also added the following in the result section: "Furthermore no difference was found for studies presented OR estimates compared to studies presented RR estimates (p for subgroup differences=1.00)."

Comment: On page 7, the authors assessed publication bias of the meta-analysis using a funnel plot, but the plot was not presented. It may be presented in the supplementary materials. Also, the authors claimed that "there was evidence of publication bias" via the funnel plot, but the P-value from the Egger's regression test was 0.596. These seem to be contradictory.

Response

We thank the reviewer for identifying the error regarding the interpretation of the funnel plot and p-value from the Egger's regression test (0.596). We have corrected the sentence: "There was no evidence of publication bias detected visually via a funnel plot, and statistically via Egger's asymmetry test (p= 0.596)".

The funnel plot for the assessment of publication bias has also added on the supplementary material as requested.

Comment: In Table 1, the author may add a column that indicates the type of effect size reported by each study (OR, RR, or HR?)

Response

We thank the reviewer for the comment. A column on Table 1 that indicates the type of effect size reported by each study has now been added as requested.

Comment: The authors presented only the funnel plot based on two subgroups of alcohol consumption (alcohol vs no alcohol or alcohol vs lowest category of exposure) in Figure 2. The funnel plots based on other classifications of subgroups may be presented in the supplementary materials.

Response

We thank the reviewer for the comment. We would be happy to supply the 7 additional funnel plots for subgroup analyses but do not feel that these would add to the paper since the results of the subgroup analyses are already presented in E-Table 2. Thus, we elect to leave it to the Associate Editor discretion whether to include these additional plots.

Comment: From Figure 2, it seems that the studies comparing alcohol vs. no alcohol were fairly homogeneous, while those comparing alcohol vs lowest category of exposure were fairly heterogeneous. The authors may investigate how differently was the “lowest category of exposure” defined in each study, because heterogeneity is a critical problem in meta-analyses and most results in this meta-analysis were highly heterogeneous (I^2 around 90%).

Response

We thank the reviewer for the comment. To clarify the heterogeneous definitions used within the studies for the “lowest category of exposure”, we have added the following descriptive information to the results section:

“However, high heterogeneity ($I^2=95%$) was detected within the second subgrouping which used the lowest category of exposure as the reference group, where the following definitions were used: no alcoholism (35, 44), no alcohol abuse (38), moderate drinking (11) ≤ 30 drinks/month(47), ≤ 0.5 bottles of vodka(39); <100 gr/day for men and <80 gr/day for women, and <20 gr/day and <10 gr/day for men and women respectively; however, the gradient of exposure did not seem to be related to the magnitude of effect”.

Minor comments:

On page 4, change “ I^2 statistics” to “ I^2 statistic”.

In Figure 3, please specify what are the dashed lines in the figure title. Also, the ranges on the x-axis in this plot are a bit confusing. Does each range (e.g., 40.1-50) indicate a single point on the x-axis or an interval from 40.1 to 50?

Response

The I^2 statistic has been corrected as requested.

We have also added the following: “On Figure 3 the dashed lines represent the 95% upper and the lower confidence intervals of the estimated relationship. Also, each range represents a single point on the x-axis”.

Reviewer: 4

Reviewer Name: Tea Kristiane Espeland Uggen Institution and Country: University of Technology Sydney, Australia Please state any competing interests: None declared

Please leave your comments for the authors below Please see comments for the authors in the attached pdf file.

Overall

You have an interesting and important research question. However, several sections of your paper are very confusing and difficult to follow. Some of the statistical tests you have performed are not explained, justified or results interpreted (only listed). There is also one occasion of contradicting findings. There are also quite a few typos and inconsistencies that need to be corrected.

Language, typos and inconsistencies

There are several typos and inconsistencies throughout the paper:

- Page 1/line 55: Full stop between last word in sentence and reference instead of space;
- Page 3/lines 3-23: Check inconsistent spacing and notation in paragraph;
- Page 3/line 39: The number “8” is written in numerical form here but in letters further down the paper. This should be kept consistent;
- Page 3/line 47: Missing space after full stop;
- Page 4/line 16: Remove space after opening bracket;
- Page 4/line 47: Missing closing bracket;
- Page 5/line 12: Missing space before opening bracket;
- Page 7/line 52-53: Inconsistent reporting of CI in this sentence compared to previous CI reporting on the same page;
- Page 7/line 33: If this is a p-value you are reporting, keep notation consistent with other p-values in your article: lower case “p”;
- Page 8/line 46: The sentence containing “... but will resulted in...” should be changed to “but would have resulted in” or “but will result in”;
- Page 8/line 49: “...did not reported...” should be “...did not report...” or “...had not reported...”;
- Page 9/line 12: Misplaced comma between “12%” and “23%”;
- Notation of decimal points keeps changing throughout the paper (x.x vs. x·x);

Response

We thank the reviewer for their comprehensive list of proposed changes. We have made the appropriate changes as requested. Regarding the following comment: “Page 3/lines 3-23: Check inconsistent spacing and notation in paragraph”, the notation used is because we followed Medical Subject Headings terms for our search in Medline and Embase. However, we share the reviewer’s concerns and to make it clearer we therefore have added in our search strategy the following: “When searching, Medical Subject Headings (MeSH) terms were used for Medline and Embase; whereas free text words were used for Web of Science”.

We have changed all the or to OR- for consistency, and have also used brackets in the correct places.

Methods

Comment: It’s not clear which form you are referring to when stating “Two reviewers (ES and JL-B) independently extracted data using a previously piloted form...”. Do you have a reference for this? (page 3/line 26)

Response

We thank the reviewer for the comment, and apologise for the confusion. We have included a blank version of the data extraction form as supplementary material.

Comment: The statement “According to CDC guidelines, we defined heavy drinking as...” is confusing – did you define heavy drinking or were they already defined in the CDC guidelines. (page 3/line38)

Response

We thank the reviewer for the comment and apologise for the confusion. To clarify this to the reader, we have added the following: “Where possible, we followed the CDC guidelines for the definition of heavy drinking.....”

Comment: You have defined levels of consumption categorically on page3/lines 34-35 as "light, moderate, heavy and alcoholism", yet on page 3/lines 39-42 the term “binge drinking” is defined in quantitative terms even though this term was not previously included as one of the four categories of consumption. Furthermore, the categories “light” and “alcoholism” are not defined quantitatively. There seems to be a lack of consistency in defining alcohol consumption. Additionally, it seems odd

that “binge drinking” and “heavy drinking” should be categorised in the same group labelled “excessive drinking”. What is the reasoning behind this? Could a person not be an infrequent consumer of alcohol but still be binge drinking on rare occasions? This section is a bit unclear. (page 3/lines 31-43)

Response

We thank the reviewer for the comments. Binge drinking has now been included in the four categories of alcohol consumption. “Also, in the main analysis, categorical measures of alcohol consumption were further defined as levels of consumption: light, moderate, heavy, **binge** and alcoholism”.

To provide some consistency across studies, where possible, we used defined consumption of alcohol based on the CDC guidelines and the Dietary Guidelines for Americans (24, 25). The aforementioned guidelines provide detailed information for the definitions of moderate drinking, binge drinking and heavy drinking. For this reason we have created categories according to these guidelines; for example the guidelines

We have amended the data extraction section of the methods to clarify this: “Where possible, we followed the CDC guidelines for the definition of heavy drinking as a weekly consumption of 15, or more drinks for men, and 8 or more drinks for women; binge drinking as 5 or more drinks during a single occasion for men, or 4 or more for women; and excessive drinking as the presence of either binge or heavy drinking (24). The Dietary Guidelines for Americans defines moderate alcohol drinking as the daily consumption of up to one drink for women and two drinks for men (25). Otherwise we accepted the definitions of alcohol that the included studies used.

Comment: In the sentence: “A score of 6, or more...” - was this cut-off value used for both cohort and case control studies (which you previously mentioned was rated out of 9) and the cross sectional studies (rated out of 7)? (page 3/line 49)

Response

We thank the reviewer for the comment and we apologise for the confusion. To overcome this issue, we have now added the following: “Two authors (ES and JL-B) independently assessed the methodological quality of the included studies using the Newcastle-Ottawa Quality Scale (26). In the process of the quality assessment of each article a maximum score of nine stars can be obtained; whereas studies with lower quality obtain fewer stars. In case of a cohort study the cohort study criteria were used; whereas for case control studies the case control criteria were used. However for a cross sectional study a modified version of the case control study criteria was used and in this case a maximum of 7 stars was given. All studies, irrespective of their design, were considered to be of high quality if they obtained a score of ≥ 6 stars”.

Comment: In the sentence: “...where the outcome measure was not assumed to be common” - what is meant by “common” in this context? This sentence is a bit confusing. (page 4/lines 8-10)

Response

We thank the reviewer for the comment. Odds ratios and risk ratios produce similar magnitudes of effect when the outcome measure has a prevalence $< \sim 10\%$. (Deeks J. Swots corner: what is an odds ratio? *Bandolier* 1996;3(3):6-7) We have clarified the text “The pooled relative risk and the 95% CI were estimated through pooling ORs and RRs together, since it was assumed that these two measures of effect would be similar due to the outcome measure being uncommon (prevalence $< \sim 10\%$).”

Results

Comment: You have listed references to the nine studies which comprised people that never consumed alcohol but not for the remaining 8 studies comprised of people that consumed the lowest quantity of alcohol. (page 5/lines 13-14)

Response

We thank the reviewer for the comment. The references of the remaining 8 studies comprised of people that consumed the lowest quantity of alcohol have been added as requested. : “The reference group for nine studies comprised people who never consumed alcohol (10, 34, 35, 37, 41, 43, 44, 46, 47); whereas the reference group for the remaining eight studies comprised people who consumed the lowest quantity of alcohol (11, 36, 38-40, 42, 45, 48).”

Comment: In the sentence: "...quality of the included studies ranged from five to eight, with a median score of six." – are these scores out of 7 or 9? You mentioned in a previous section that some were out of 7 and some out of 9. You may want to mention what you are comparing these scores to again. (page 5/lines 21-22)

Response

We thank the reviewer for the comment and again we apologise for the confusion. We hope that we have clarified it in our above response. To further clarify it to the reader, we have now added the following: “The methodological quality of the case control, cohort and cross sectional studies ranged from five to eight, with a median score of six. Ten studies were deemed to be of high quality (>6 score). The results of the quality assessment are presented in detail in Table 2”.

Comment: The overall I2 is very high (91%) and you should provide interpretations of the effects of this. (page 7/line 32)

Response

We thank the reviewer for this comment. However as the reviewer will notice we have mentioned in the meta-analysis findings that we have performed subgroup analyses as an attempt to identify possible reasons of the high level of heterogeneity. “Subgroup analyses exploring the reason for heterogeneity in the meta-analysis of these 14 studies are presented in the Supplementary material (see Table E2). Heterogeneity was not explained by study design (case control, longitudinal/cohort, cross sectional; p for subgroup differences=0.07), methodological quality (high versus low; p=0.09) or gender (male versus female; p=0.74). However, significant differences were found according to adjustment for confounders (adjusted versus unadjusted; p=0.03), continent of study (America, Europe, Australia; p=0.0003), and ascertainment of CAP (clinical diagnosis vs death records; p=0.002). Furthermore no difference was found for studies presented OR estimates compared to studies presented RR estimates (p for subgroup differences=1.00)”.

Furthermore we have stated in the strengths and limitation section the following: “There was significant heterogeneity between the studies in our analysis, but our subgroup analyses indicate that this arose primarily from the continent in which the study was carried out (America, Europe, Australia); adjustment for confounders; and the ascertainment of CAP (death vs clinical diagnosis)”.

Comment: There are contradicting findings with the statement in “RESULTS”: “There was evidence of publication bias detected visually via a funnel plot, and statistically via Egger’s asymmetry test (P=0.596)” on page 7/line 32-33 and with the statement in “DISCUSSION”: “...with no evidence of publication bias” on page 8/line 34-35. Furthermore, the p-value listed seems to be inconsistent with the conclusion drawn in the results. (page 7/lines 32-33)

Response

We thank the reviewer for identifying this error which was also raised by Reviewer 3. We have therefore corrected that: “There was no evidence of publication bias detected visually via a funnel plot, and statistically via Egger’s asymmetry test (p= 0.596)”.

Comment: You should provide the funnel plot described on page 7/line 32-33 as “There was evidence of publication bias detected visually via a funnel plot, and statistically via Egger’s asymmetry test”. (page 7/lines 32-33)

Response

We thank the reviewer for this comment which was also raised by reviewer 3. The funnel plot for the assessment of publication bias has added on the supplementary material as requested.

Discussion

Comment: What is meant by “relatively high intakes”? You previously categorised alcohol consumption to be “low/moderate/high/alcoholism”. Which ones are “relatively high”? Could you also provide a quantitative measure for this term? (page 8/lines 33-34)

Response

We thank the reviewer for the comment. We have amended the sentence in the discussion to: “The effect was strong, with a 1.8 fold increase in risk among those who consumed alcohol at all, or in higher amounts, relative to those who consumed no, or lower amounts of alcohol respectively and significantly related to level of intake, with no evidence of publication bias”.

The specific measures of exposure to alcohol extracted from the studies and used within the analyses are already presented within Table 1. We have elected to not provide additional information which describing these in the text of the results section since we think this would not significantly add to the paper. However, we would be happy to provide this additional descriptive information if the Associate Editor would prefer.

Comment: The concluding statement is a bit of an over-reach given not only the term “relatively high intakes of alcohol” and the high I2 but also the conclusion drawn on morbidity and mortality. (page 9/lines 28-31)

Response

We thank the reviewer, and have amended the concluding sentence to reflect their concerns: “Our findings thus provide clear evidence that alcohol increases the risk of pneumonia. Informing people who drink alcohol of this risk, especially those who consume high levels of alcohol, both in clinical contacts and through public health policy, may therefore help to prevent this disease”.

VERSION 2 – REVIEW

REVIEWER	Lifeng Lin Florida State University
REVIEW RETURNED	31-May-2018

GENERAL COMMENTS	The authors have satisfactorily address my concerns, and I do not have further comments.
--

REVIEWER	Francisco Sanz Herrero Pulmonology Department, Consorci Hospital General Universitari de Valencia, Valencia (Spain)
REVIEW RETURNED	05-Jun-2018

GENERAL COMMENTS	Thanks for the changes and improvement of the reviewed manuscript.
--